# Effect of Processing on Phenolic Composition of Olive Oil Products and Olive Mill By-Products and Possibilities for Enhancement of Sustainable Processes

**Fereshteh Safarzadeh Markhali** 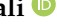

CEB—Centre of Biological Engineering, Campus of Gualtar, University of Minho, 4710-057 Braga, Portugal; id7987@alunos.uminho.pt or Fereshteh.safarzad@graduate.curtin.edu.au

**Abstract:** The bio-functional properties of olive oil products and by-products rely greatly on the proportions and types of the endogenous phenolics that may favorably/unfavorably change during various processing conditions. The olive oil industrial activities typically produce (i) olive oils, the main/marketable products, and (ii) olive mill by-products. The mechanical processing of olive oil extraction is making progress in some areas. However, the challenges inherent in the existing system, taking into consideration, the susceptibilities of phenolics and their biosynthetic variations during processing, hamper efforts to ascertain an ideal approach. The proposed innovative means, such as inclusion of emerging technologies in extraction system, show potential for sustainable development of olive oil processing. Another crucial factor, together with the technological advancements of olive oil extraction, is the valorization of olive mill by-products that are presently underused while having great potential for extended/high-value applications. A sustainable re-utilization of these valuable by-products helps contribute to (i) food and nutrition security and (ii) economic and environmental sustainability. This review discusses typical processing factors responsible for the fate of endogenous phenolics in olive oil products/by-products and provides an overview of the possibilities for the sustainable processing to (i) produce phenolic-rich olive oil and (ii) optimally valorize the by-products.

**Keywords:** processing; phenolics; olive oil products; olive mill by-products; sustainability; extraction



## 1. Introduction

Polyphenols, the secondary metabolites and the predominant groups of phytonutrients in plants, are highly valued for their bio-functional properties and defense mechanisms. Among other benefits, they confer antioxidative [1] and antimicrobial activities [2,3]. Of all the plant-derived foodstuff, the unique healthful and organoleptic attributes of olive oil products have noticeably caught the global attention, signifying their distinctive chemical and molecular characteristics [4]. In this regard, the agro-industrial activities of olive crops, which have been traditionally and economically substantial particularly in the Mediterranean region, have progressively increased primarily for olive oil production [5,6]. Olive oil, on top of being a good source of unsaturated fatty acids (around 72% monounsaturated fatty acids primarily oleic acids, and 14% polyunsaturated fatty acids [7]), is highly valued as a source of minor bioactive compounds including phenolics [8,9]. The bio-functional potential of phenolics, however, relies greatly on their proportion, molecular structure/interactions, and chemical metabolism [10]. In the olive oil industry, there are different classes of olive oil products, typically characterized by the specified/standard quality parameters. The main groups, among others, include extra virgin olive oil (EVOO), virgin olive oil (VOO), and refined olive oil [7]. Of all designated groups, EVOO is known as the highest quality, containing total polyphenols in the range of 50 to 1000 mg/Kg, and acidity level below 0.8 g/100 g [11].

The functional potency of olive oil partly relies on the quantities of specific groups of endogenous phenols [12] that are increased/decreased depending on the various pre/post-harvest activities. In this respect, there has been a growing research trend towards the exploration of the fate of endogenous bioactive molecules, particularly polyphenols, in olive oil and its by-products, through various agro-industrial steps. Together with other factors, the industrial processing and storage conditions are highly influential in the degradation/formation of polyphenols [11,13]. The initial point of the optimum preservation of endogenous phytonutrients relies on a good handling of raw materials that may be achievable through a minimum/careful storage time. Ideally, the storage time for the harvested crops to be processed for olive oil extraction should not exceed 48 h [14]. The prolonged storage of harvested olives may give rise to spoilage (oxidative and/or hydrolytic) in the final product [14]. Together with the storage factors, the processing parameters during olive oil extraction, particularly malaxation temperature/time, plays crucial roles in the phenolic composition of the extracted oil, which is further outlined in more details in Section 2.

Olive tree farming, in addition to the target/marketable products, generates a range of waste stream/by-products (Figure 1) that is considered to be a good source of bio-functional molecules but they are currently underused for limited/low-value applications, signifying the inefficiency of the processing system currently used for their extraction/recovery.

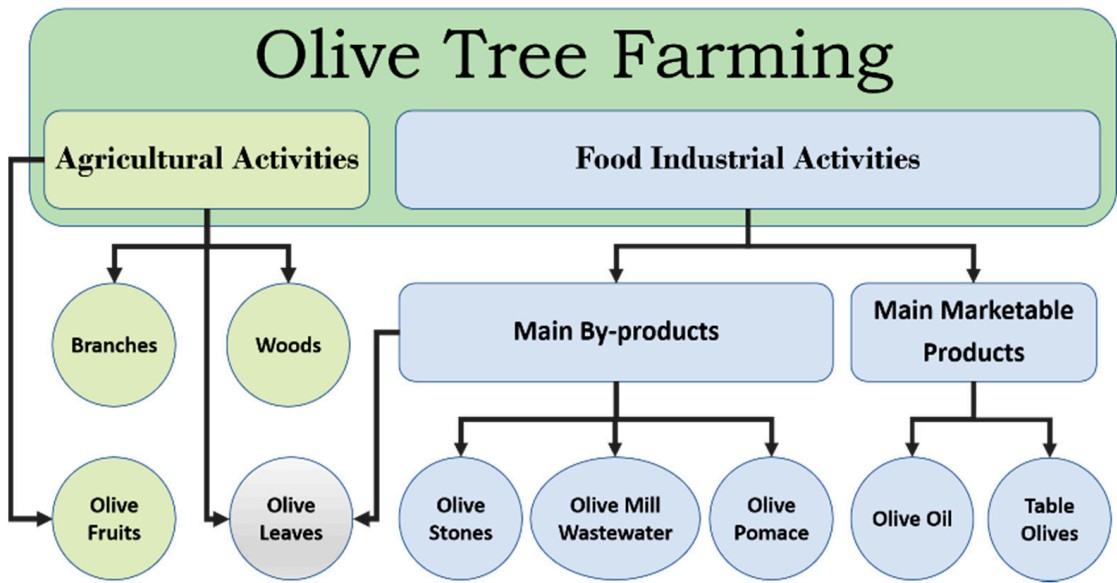

**Figure 1.** Major products and by-products produced from agro-industrial activities of olive fruits.

Green technologies show potential for delivering optimal processing of olive oil products, and sustainable valorization of olive mill by-products, Although, there is a trend towards the processing developments in olive oil industry but, having said that, the proposed emerging methods have not quite reached the production system. The existing mechanical operations commonly used in both olive oil production and by-product re-utilization often represent (i) less efficiency of the processing of the main products, and (ii) low value applications of the processed by-products. This review describes the role of the processing on the fate of major phenolics in olive oil products and by-products and emphasizes the possibilities for a sustainable processing system that may enable achievement of the increased recovery of health-promoting phenols in olive oil, as well as improved valorization of olive mill by-products/residues that have the potential for high-value applications across various industrial sectors.

## 2. Olive Oil Products

*Overview of Characteristics of Typical Endogenous Phenolics and the Effect of Processing Conditions on Their Loss/Gain*

The phenolic constituents of olive oil typically include oleocanthal, oleacein, oleuropein, hydroxytyrosol, and tyrosol. Of all endogenous phenolic groups, oleocanthal (decarboxymethyl ligstroside aglycone), a representative secoiridoid, is considered distinctively beneficial, particularly with respect to its anti-inflammatory effects, which are comparable to those exerted by ibuprofens [15–18]. Oleocanthal potentially plays a part in curtailing the activities of inflammation-induced enzymes such as cyclooxygenase-2 (COX-2) [15,18–20]. A daily consumption of 50 g of extra virgin olive oil, which typically contains around 10 g oleocanthal, is reportedly proportionate to 10% of the low/daily dosage of ibuprofen recommended for adult consumption to treat soreness [9]. Moreover, like other COX-2 inhibitors, it has been suggested that the regular consumption of this natural ingredient competitively confers protection against chronic disorders including cardiovascular and some types of carcinogenic diseases [9]. Further, this healthful ingredient is potentially inhibitory against progressive brain disorders such as Alzheimer's disease [21].

Another representative secoiridoid in olive oil is oleacein (3,4-dihydroxyphenylethanol-elenolic acid dialdehyde), which possesses a molecular structure relatively similar to that in oleocanthal but slightly differs in terms of the hydroxyl group, where oleacein bears one further OH group [17]. This component significantly exhibits antioxidation [17] and anti-inflammation [22], and it reportedly contributes to the suppression of platelet aggregation, partly through hindering the action of 5-lipoxygenase (5-LOX) enzymes [8]. The antioxidative activity exerted by oleacein has been justified in the literature in a way that it may competitively be greater than that exhibited by hydroxytyrosol and oleuropein [16,23]. From the perspective of sensory attributes in extra virgin olive oil, oleacein contributes more to the bitterness [17] rather than the pungency/peppery sensation that is more attributable to oleocanthal [9]. Oleacein generally manifests low storage stability and has less heat stability compared to oleocanthal [17].

In the literature, much research has been carried out to re-evaluate various processing conditions to optimize recovery of phytonutrients in the extracted oil. The proportions and discrepancies of intrinsic phenolic constituents rely partially on the mechanical operations used for the extraction of olive oil [24]. The extraction of olive oil primarily involves the exertion of oil release that is predominantly present in the mesocarp and to a lesser extent in endocarp of the olive fruits [25]. An optimum oil extraction may be achieved following (i) disintegration of the cell membrane of mesocarp through the crushing step, and (ii) progression of the oil droplet coalescence through the malaxation (mixing) step [25,26]. A typical example highlighting the significant roles of the processing in phenolic composition is the variability of oleocanthal concentrations among various extra virgin olive oils, which can be as little as 0.2 mg/kg or as great as 498 mg/kg oil [9]. Cicerale et al. [9], through their experiment of olive oil extraction where both fruits and pits were crushed during malaxation, observed lower concentration of oleocanthal (around 43.8 mg/kg oil). The authors reported that the extracted oil through the crushing of de-stoned pulp represented higher percentage of oleocanthal (around 54.8 mg/kg oil). The pressure of crushed stones in olive paste may unfavorably cause peroxidase activity and oxidative damage, which brings about reduction of the phenolic proportion in the extracted olive oil [27]. Oleocanthal is also susceptible to the light exposure, which occurs often during storage [17].

The proportion of each phenolic compound also varies in different classes of olive oil. Hydroxytyrosol, a phenylethanoid, yields considerably greater in EVOO compared to those in refined olive oil, representing around 14.32 and 1.74 mg/kg oil, respectively [28]. Hydroxytyrosol, even being present in low concentration in olive oil [29], is highly prized for its pronounced bioactive effects and exhibits relatively high oxidative stability in virgin olive oil [30], while some other groups of endogenous antioxidants such as tyrosol and ligstroside aglycone exhibit less potency in this respect [14]. The antioxidative potency

of hydroxytyrosol has also been reported to be greater than that exerted by butylated hydroxytoluene (BHT) [31], which further merits its utilization as a natural antioxidant alternative. The bioconversion of tyrosol into hydroxytyrosol has been studied by numerous researchers. Bouallagui and Sayadi [32] performed research on the catalyzing influence of the whole strain of bacterial cells of *Pseudomonas aeruginosa*, grown on tyrosol using a fermenter, on the conversion of tyrosol to hydroxytyrosl, finding 86.9% bioconversion recovery. Azabou et al. [33] observed optimized recovery of hydroxytyrosol (64.36%) through using photocatalytic oxidation of tyrosol.

The industrial processing of olive oil commonly uses thermal application during malaxation of olive paste, which is effective in gaining an improved extraction of oil from olive pulp through the action of coalescence of oil droplets [13]. The proportion of phenolics in olive oil partly reflects the rate of their solubilization and chemical/enzymatic reactions, which are highly affected by the processing conditions including malaxation parameters [13]. Chemical alterations occur when olive paste is in contact with the air during malaxation, which may adversely develop degradation/oxidation of aglycones (the non-sugar groups of glycosides), typically by enzymatic activities [14].

The increased malaxation temperature may foster catalyzing reactions of some oxidative enzymes such as polyphenol oxidase and peroxidase [13]. The use of heating during mixing process may be favorable or detrimental to the resulting phytonutrients of the end products. Some phenolics such as oleocanthal are relatively heat-stable, and this is particularly the case when significant concentration of oleocanthal is present at the early stage of recovered extra virgin olive oil [17]. de Torres et al. [34] found increased proportion of oleocanthal (289.4 mg/kg), as well as hydroxytyrosol (7.83 mg/kg), with the rise of temperature of up to 60 °C. By contrast, concentrations of phenolics such as oleuropein, ligstroside aglycone [35], luteolin, and flavonoids [34] have shown reverse correlation with the temperature increase.

The malaxation time comparably is a decisive factor for phenolic status in the oil [13]. Gomez-Rico et al. [36] observed a 70% increase of C6 aldehydes (a distinctive volatile component of olive oil [37]), predominantly *E*-2-hexenal (that is characterized by green leaf and apple sensory attributes), when the malaxation time was exceeded from 30 to 90 min. Using an extended kneading time provides an opportunity for the increased dispersion of the oily phase and liberation of the volatile compounds [36]. The study of Miho et al. [38] demonstrated an increased concentration of oleacein and oleocanthal in virgin olive oil with the increase of malaxation time. However, these authors reported a noticeable reduction of isomers of oleuropein aglycone (3,4-DHPEA-EA) and ligstroside aglycone (p-HPEA-EA) in the oil after using a prolonged malaxation. Likewise, total phenolic concentration is reportedly reduced within lengthy malaxation time [13].

The main processing steps involved in olive oil production consist of cleaning of harvested olives, crushing, malaxing, and phase separating. The separation of olive oil uses mechanical operations either based on (i) the conventional approach, using discontinuous pressing system, or (ii) the modern approach, using continuous centrifugation system [25,39]. Table 1 provides a comparison between the main separating techniques often used for the extraction of olive oil. The conventional approach is typically equipped with millstones that favorably use less crushing time via gentle/low-speed spinning, and thus the incidence of emulsion is potentially impeded, enabling improved coalescence and increased oil recovery [25]. However, together with other downsides inherent in the labor intensity and inefficient system operation, including low working capacity, include the potential contamination of the filter mats and the need for a strict hygiene routine [25]. The modern centrifugation system often uses either a two-phase or a three-phase decantation [25,40,41]. The system commonly employs metal crushers that help increase extractability of the oil and recovery of total phenolics [25]. However, the emulsion intensity and development of bitter taste in the extracted oil are of typical disadvantage. In general, application of the two-phase decanters in place of a three-phase centrifugation

system eliminates/minimizes the use of water addition to olive paste and enables obtaining an increased recovery of polyphenols in the extracted oil [41–43].

**Table 1.** Comparison of phase separating techniques for olive oil extraction.

| Phase Separation Method *Principle and Outputs* | | Benefits | Downsides | References |
|---|---|---|---|---|
| **Conventional Approach** *(Discontinuous system)* | **Pressing** *(using millstones)* *Outputs:* Pomace Olive oil Wastewater | Less crushing time Less emulsion Improved coalescence Increased oil recovery Low energy consumption Low moisture content pomace | Labor intensity Discontinuous/inefficient system Low working load Needs strict hygienic routines | [5] [25] [44] |
| **Modern Approach** *(Continuous centrifugation decanting)* | **Two-phase** *Outputs*: Wet pomace Olive oil | Eliminates use of water Greater phenolic recovery High quality/yield of oil Less use of energy No generation of wastewater | High moisture content pomace Lower working load Organoleptic acceptance *(too bitter/pungent)* | [25] [39] [41] |
| | **Three-phase** *(Addition of water)* *Outputs:* Olive cake Olive oil Wastewater | High working load Automated system Less labor/production cost High quality/recovery of oil Moderate moisture content pomace | Need additional use of water and energy Generation of wastewater Lower phenolic recovery Wastewater management cost | [25] [39] |

Furthermore, other accountable factors such as the appropriateness of analytical methods selected for the quantification of endogenous phenols play a decisive part in the precision and reliability of the obtained results. Application of high-performance liquid chromatography (HPLC), as an official method to measure individual phenolics of olive oil, uses polar solvents including methanol and water. This method may not be an ideal approach for accurate analysis of dialdehyde phenols including oleocanthal, oleacein, and derivatives of hydroxytorosl and tyrosol due to their potential reactions with the above-mentioned solvents, giving rise to peak broadening and possible developments of hemiacetal and acetal derivatives [45–48]. However, the research of de Medina et al. [16] examined the possibility of conversion of oleocanthal and oleacein into hemiacetal and acetal derivatives through liquid chromatography with tandem mass spectrometry (LC–MS/MS) and found that the methanol/water had no/little effects on the formation of these artifacts. Indeed, a slight proportion was detected when methanol gradients were applied under acidic conditions at the stage of the chromatographic separation. Their investigation postulated the use of acetonitrile for the extraction step and methanol-based gradients for the chromatographic separation step as a suitable approach for measuring oleocanthal and oleacein in olive oil.

Further, nuclear magnetic resonance (NMR) spectroscopy, using deuterated solvents, has been viewed as an ideal analytical method for accurate measurements of secoiridoid aldehydes [13,45,47]. The use of NMR has shown effectiveness and selectivity in the detection of some isomeric aglycones that may not be detectable through a normal and/or reversed-phase chromatography as they inherently transform to other types of aglycone isomers. This is potentially indicative of the interaction of extracted oil with the silica-based stationary phase [13,49]. Diamantakos et al. [13], in their research on investigation of some key factors (including malaxation time/temperature) on the concentrations of phenolics of EVOO from various cultivars, used NMR for the quantification of secoiridoid phenolics. Together with oleocanthal and oleacein, other major secoiridoid derivatives including oleomissional and oleokoronal were detected and quantified. These compounds are the recently known isoforms associated with oleuropein aglycone and ligstroside aglycons which are well detected by means of NMR method [13,49]. In this regard, it is essential

to employ appropriate analytical techniques for phenolic measurements because using inapplicable methods may lead to misinterpretation of phenolic proportions in olive oil.

## 3. By-Products of Olive Oil Industry

The growing industrial production of olive oil that is partly in response to the increased global desire [41], has resulted in a massive generation of a varied range of by-products/waste streams (Figure 1). Among others, they include liquid waste (olive mill wastewater), and solid waste (olive pomace) [41]. The discharge of these processing biomass, particularly the liquid effluent, represents (i) environmental impact, causing toxicity, contamination, and pollution [41,50], and (ii) economic damage to the respective manufacturers [51].

### 3.1. Functional Potential and Processing Considerations for Extraction

The bioactive potency of natural phenols present in olive mill by-products has been well reviewed in the literature. Their phenolic concentration is reportedly much greater than those remain in olive oil products, representing around 98% and 2%, respectively [52]. Their exploitation not only helps address the environmental issue but is of benefit to provide natural bio-ingredients with value addition potentials that enable a sustainable re-use for food or non-food applications [53]. As an example, research has demonstrated the favorable effectiveness of incorporating the extracted phenols such as hydroxytyrosol and oleuropein into other food products such as vegetable oils to promote their functional/nutritional properties [51]. However, the achievement of a sustainable exploitation system is highly dependent on the appropriateness of the mechanical extraction techniques [53]. This is partially because the optimum recovery of target phenolics may be hampered due to (i) complexity of molecular structure of phenolics, as they are often attached to glycone (sugar) or protein groups, and (ii) variability of biochemical pathway and the possibility of unfavorable formation of some phenolic derivatives that may impede the optimum recovery of desired compounds [54].

The traditional types of phenolic extraction include Soxhlet, hydro-distillation [51], and solvent extraction methods [52]. These methods are still widely used in the agro-industrial system partly because of their simplicity, flexibility, and versatility [54]. Application of innovative technologies, which are somewhat being adopted in some areas, potentially enables the achievement of (i) operation efficiency [55], (ii) improved quality, (iii) productivity with lower cost, and (iv) environmental sustainability [54,55]. Among the key processing paraments of extraction include extraction time/temperature and solvent type/ratio [54]. Together with other disadvantages of the conventional system inherent in the need for prolonged extraction time and/or high temperature is the need for using relatively high proportion of solvents (such as methanol and acetone). To date, numerous emerging methods such as microwave-assisted extraction (MAE) [51], ultrasound-assisted extraction (UAE) [51,54], infrared-assisted extraction (IR-AE), membrane separation, and supercritical fluid extraction (SFE) [51] have been recommended to overcome some of the challenges associated with the conventional/existing methods. A typical advantage, among other things, includes using safe/organic solvents such as supercritical carbon dioxide, water, and ethanol [56].

The suitability of the extraction system to enable optimum recovery of the given compounds also relies on the solubility and polarity of target molecules [56]. For example, some organic solvents such as hexane may not be applicable for the extraction of polar phenolics, producing poor solubility and low yield recovery [56]. Moreover, using a single organic solvent may act inadequately on the efficiency of the diffusion rate/mass transfer of polar compounds. To tackle this hurdle, researchers have suggested some alternatives, such as using selective solvents proportionally, e.g., water mixed with ethanol or acetone [54]. The functional potential and processing considerations for the main types of olive oil by-products are outlined as follows:

### 3.1.1. Olive Leaves

Olive leave by-products are not only accumulated during the agricultural/pruning activities but are massively generated through the industrial activities of olive oil production, which account for 5% [57] up to 10% [58] of overall weight of olives harvested for processing. These biomass residues are presently underexploited and usually being re-used as animal feed [59,60] while having a great potential for high-value addition owing to their good source of bioactive compounds.

Olive leaves are known to be markedly rich in oleuropein (a phenolic secoiridoid) in part because they do not undergo the extraction processing of olive oil. Indeed, they are typically removed at the preliminary stage of olive oil extraction, prior to the milling/crushing operations. In this regard, oleuropein constituents that are liable to be degraded/hydrolyzed during oil extraction potentially remain intact in olive leaves [61]. The presence of valuable natural phytonutrients such as oleuropein (around 14% dry basis) and oleanolic acid (around 3% dry basis) in olive leaves [57] has currently led to the extensive research studies, with particular attention to the design formulation of extraction methods/parameters to ideally liberate the antioxidants of interest [62]. During an ideal extraction process, the release of functional molecules is exerted, which may favorably increase the chance of their bioavailability. For example, the incidence of decomposition/hydrolysis of oleuropein molecules that gives rise to production of hydroxytyrosol and elenolic acid can be favorable if the intention is to liberate hydroxytyrosol. The degradation of oleuropein, together in presence of acids and metal ions, may occur through enzymatic reactions and high temperature [63]. These phenomena can be undesirable particularly if the main purpose of the extraction is to isolate oleuropein. On this account, the extraction parameters need to be designed cautiously to ensure no/minimum detrimental effect incurs on the molecular structure of the target compounds.

Devising sustainable extraction techniques is of paramount significance, particularly when considering the susceptibility of phenols to high temperature [60] and oxygen [64]. The inclusion of appropriate pre-treatments, such as blanching through olive leave extraction, has been found to be influential in obtaining increased recovery of phenolics. Zeitoun et al. [65] observed improved recovery of total phenolic compounds, up to about 61.70% when the leaves were subjected to blanching using hot water at 90 °C for 20 min. The types of sample preparation and storage parameters of leave samples prior to the extraction process are also accountable to the level of depletion/recovery of the representative phenolics, such as oleuropein and verbascoside. Malik and Bradford [66] carried out an investigation on various processing parameters on the loss/preservation of phenolics in the extracted olive leaves and found significant efficiency of recovery of oleuropein and verbacoside when fresh leave samples were dried at 25 °C, whereas drying of leaves at elevated temperature (60 °C) lowered the concentration of total phenolic compounds. The authors of this study also observed that defrosting frozen olive leaves within 5 min and 2 min lowered the recovery of oleuropein up to 57.7% and 53.5%, respectively.

### 3.1.2. Olive Pomace

Olive pomace, namely, olive cake, refers to the residual solid by-product that is made up of olive pulp (up to 90%) and olive stones [67,68], which remains after processing of olive oil extraction through centrifugation or pressing [6,69]. The solid residue that is produced from a two-phase centrifugation system is known as "two-phase olive pomace", also termed olive mill solid waste, which contains around 65% [70], up to 70% moisture [68]. The pomace generated from a three-phase decanter contains a lower amount of water, around 45% moisture [68].

Olive pomace is abundant in an array of phenolic compounds [71], including hydroxytyrosol (around 1.8% [68]), oleuropein, verbascoside, and tyrosol [6], making it a valuable candidate for bio-functional and value-added applications. Many studies have been carried out on the processing design (including extraction time, temperature, and solvent types) to optimally extract nutritive components of this by-product. Vitali Čepo et al. [72] observed

efficiency of using ethanol (60%) in the extraction of total phenolics (3.62 mg gallic acid equivalent/g pomace) and oleuropein (115.14 mg/kg pomace) through the extraction temperature at 70 °C for 2 h. The use of methanol as the extraction solvent has also been reported to notably facilitate the extraction yield of total phenolics from olive cake, as evidenced through the extraction process (i) at 70 °C for 12 h using 80% methanol [52], and (ii) at 70 °C for 3 h using 40% methanol to extract olive pomace [73].

The lipid/oily phase of olive pomace is commonly extracted using solvents and subjected to a refining process to make it edible, where it is then mixed with virgin olive oil (around 5% to improve its quality/sensory attributes), being commercially known as refined olive pomace oil [69]. It is crucial to deliver these processed residues promptly to the pomace oil manufacturing to prevent/minimize possibility of oxidation/rancidity in the final oil product.

### 3.1.3. Olive Mill Wastewater

The generation of olive mill wastewater (OMWW) as the liquid effluent occurs particularly through a three-phase centrifugation system that accounts for a large amount, about 50% of the total yield of process output, after each extraction process [5]. The endogenous polyphenols in OMWW (ranging from 0.5 to 24.0 g/L wastewater [60]) may significantly exhibit health benefits such as antiradical and antimicrobial activities [6]. The recovery of bio-phenols from OMWW is achievable through a variety of extraction methods. Solvent extraction technique is a relatively more commonly used method but, due to its drawback inherent in the need for a sizable portion of solvents, can be ideally replaced by a supercritical extraction system (although it comes with capital/apparatus expenditure) [74]. The membrane filtration method is considered as a potentially advantageous technique for valorization of OMWW, which, among others, include reducing energy use and eliminating additive use [74]. The main classifications, other than the conventional membrane technique, include nano-filtration, ultrafiltration, microfiltration, and reverse osmosis membranes that are regarded as highly effective means, enabling applicable recovery and isolation of target molecules [74]. Zagklis et al. [75] performed a study on the extraction of phenolics from OMWW—using membrane filtration (comprising reverse osmosis concentrate, nanofiltration, and ultrafiltration), and isolation of the recovered compounds—using resin adsorption/desorption. In their research, the concentrated phenolic compounds represented 378 g gallic acid equivalent per liter as compared to those in the original/non-filtered OMWW (2.64 g/L). As described in Section 3.1.1, the biosynthesis of hydroxytyrosol may come about when oleuropein is hydrolyzed/decomposed during processing of oil extraction, and it is largely accumulated in olive mill wastewater. Hydroxytyrosol is highly treasured for its bio-functional qualities and has the market potential/industrial demand in the food and dietary system, particularly when considering the expensive and complicated process to synthesize this component [32,60].

Olive oil industries (particularly in Spain) continue to adopt the replacement of the three-phase centrifuge decanter by the two-phase system (namely ecological) that generates a significantly lower amount of OMWW [53,70,76]. A sustainable reutilization of OMWW provides a good marketing potential for value addition/nutraceutical applications in nutrition and food system, particularly when a feasible methodology is designed to gain optimum extraction yield. Optimization of process design that serves both effluent treatment and valorization of OMWW has been the topic of research studies, and some proposed methods are outlined in this review (Section 4).

### 3.1.4. Olive Stones

Olive Stones, namely, the pits, refer to the endocarps of olive fruits [77] that account for around 10% [53] or up to 27% of total weight of fruit [78]. Olive stones and/or their kernel (the seeds surrounded by endocarps) are a good source of phenolic compounds with bio-functional potentials [78]. Among them are hydroxytyrosol (0.4–1.9 g/100 g of whole stone db [79]), tyrosol (0.1–0.8 g/100 g of whole stone db [79,80]), and oleuropein (0.1–0.2 g/100 g

of whole stone db) [79]. Specific types of phenolics such as verbascoside are only present in the kernel part of the stone (0.4–0.8 g/100 g dry kernel) [79]. Elbir et al. [81] in their experiment reported various concentrations of total phenolics as follows: 11.32, 4.55, and 3.56 mg gallic acid equivalent per gram dry basis of olive stones extracted from Moroccan olives of Picholine, Haouzia, and Menara cultivars, respectively.

Olive stones as the natural lignocellulosic biomass are also a weighty source of polymers including lignin and cellulose that are valued for direct/indirect fields of applications such as combustion for bioenergy uses [79]. The ability to devise an applicable process formulation via innovative means, promptly affects the ability to re-use the whole olive stones and convert the desired components into high-added value products sustainably. For instance, using pre-treatment via a steam explosion prior to the isolation and fractionation of target biomolecules helps ensure isolation of components, which is otherwise likely to be hurdled due to the physical and chemical characteristics of these products [79].

## 4. Sustainable Processing System in Olive Oil Industry—An Overview of Optimizing Value

### 4.1. Olive Oil

In the olive oil industry, there is a trend towards evolving the processing technologies to enhance both quality and extraction yield. A typical example is the replacement of the three-phase centrifugation system by the two-phase decanters, which is reportedly more effective in the increased concentration of total phenolics as well as individual/specific phenolic groups such as oleuropein aglycone, hydroxytyrosol, and tyrosol. However, the current advancements of the processing technologies used in olive oil production may not be sufficient, although virgin olive oil is endogenously abundant in bioactive compounds; however, given its richness of unsaturated fatty acids [82], the oxidation of oil during processing/storage, e.g., as a result of prolonged storage [82] and light exposure [82,83], is often expected, which in turn brings about the degradation of endogenous phenolics. Furthermore, some phenolics are inherently more liable to be diminished/degraded through various steps of processing, e.g., decomposition of oleuropein when the temperature is increased. Research studies have suggested alternative means to overcome oxidative deterioration of olive oil. As an example, the addition of olive leaves in advance of crushing step to enhance olive oil quality. The isolated phenols from olive leave by-products can be generally used as propitious value-added products through their incorporation into lipid-based foodstuff with poor oxidative stability such as refined olive oil [84–86].

In general, each group of phenolic components is influenced differently during the processing of olive oil, and this may hamper identification of a decisive pattern for the synthetic routes/formation of each class of phenols during different processing conditions [13]. Although parameters such as malaxation temperature may favorably correlate with the increase of some phenolics/total phenolics, but this may not be applicable to the production of virgin olive oil in terms of the sensory acceptability that is likely to be adversely affected by the rise of temperature [34,87]. Another study reported that the rise of temperature may be favorable in terms of yield improvement, but this can bring about deterioration/oxidative rancidity [88] with resulting impact on the phenolic profile/potency. Possible solutions to upgrade the conventional malaxation operation are (i) inclusion of chemical aids such as pectolytic and cellulolytic enzymes that may assist in rupturing the cellular structure and liberating the phenols into the oil fraction [6,89], and (ii) olive paste treatment using emerging technologies during malaxation to complement the extraction efficiency. Puértolas and de Marañón [89] through their investigation on the treatment of olive paste using pulsed electric field during malaxation of olive oil, reported an increased recovery of total phenolics (11.5%) as well as yield percentage of the extracted olive oil (13.3%) when compared to the samples without treatment. Other types of emerging techniques, with the potential to overcome the challenges inherent in the malaxation operation, include microwave heating, ultrasound technologies [90,91], and high-pressure processing [91].

### 4.2. Olive Mill by-Products

The enormous generation of olive mill by-products has prompted the idea among scientists to propose green solutions for high-value applications to optimally recover and re-use these valuable substances and enable sustainable marketability in food/dietary and non-food system [76]. Currently olive mill by-products have found rather low/moderate-value applications for direct/indirect uses in the agriculture and industrial systems [92]. Some applications use de-fatted pomace [25,93] for natural renewable energy sources, via thermochemical decomposition of organic compounds (such as cellulose and lignin), through pyrolysis and combustion, as well as gasification [67]. Other uses have reached as far as animal feed and compost [92], and to some extent, have found applications in the pharmaceutical and cosmetic sectors but they are predominantly under-exploited while being valuable as health-giving products [53].

The ability to deliver the high-value applications of olive mill by-products entails consideration of numerous factors associated with the green valorization process, which partly highlights the importance of (i) the selected extraction method, with special attention to gaining higher recovery of bio-functional compounds, as well as their conversion into value added ones; (ii) the challenges involved in meeting the sustainability criteria; and (iii) high-value applicability of the recovered products in the industrial sectors to increase marketing opportunities. An overview of the typical factors involved in a green valorization of residual biomass generated in the olive oil industry is illustrated in Figure 2.

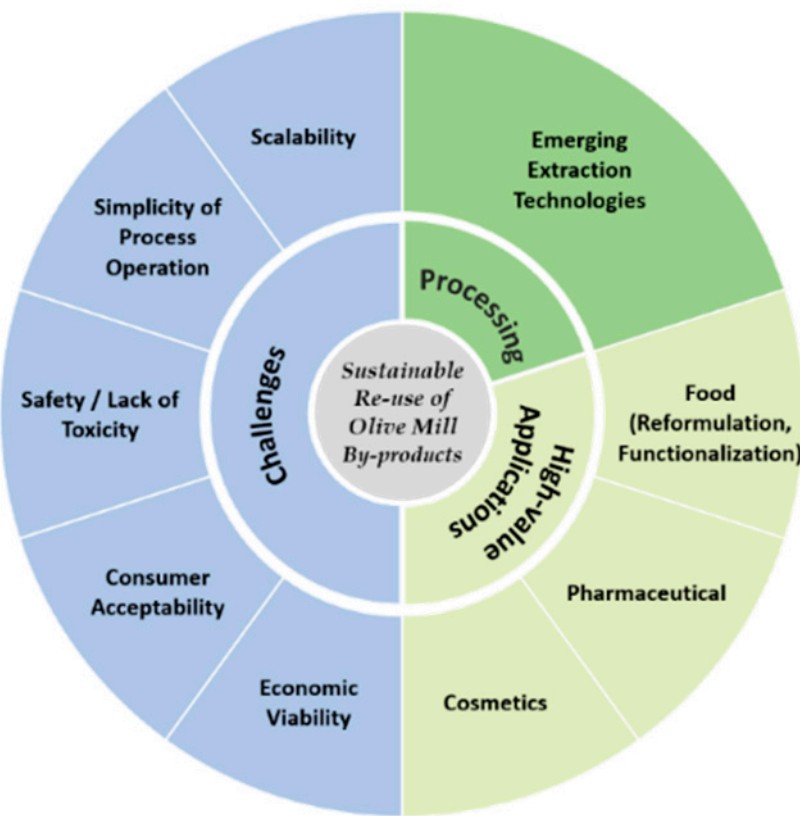

**Figure 2.** Summary of the associated factors accountable for the sustainable valorization of olive mill by-products.

The treatment of olive mill wastewater comes with a great challenge due to its phytotoxicity [5,94]. Numerous waste management strategies have been recommended in this respect, most of which favorably enable the liberation of bioactive molecules. Sygouni et al. [70] postulated the effectiveness of membrane filtration as an ideal eco-friendly method for the treatment of OMWW as well as the achievement of phenolic

recovery. Other methods to achieve purification of functional ingredients from OMWW include solvent extraction (widely used), chromatographic, and enzymatic-assisted extraction methods [95].

Kachouri and Hamdi [96] justified the influence of incorporation of olive mill wastewater (fermented by *Lactobacillus plantarum*) to olive oil that facilitated the decrease of phenolics in wastewater residue and increase of phenolics in olive oil. This is primarily attributed to the abilities of *Lactobacillus plantarum* to de-polymerize high-molecular-weight phenolics in OMWW that potentially enables their movement from wastewater to the oil [96]. These authors found a significant increase of polyphenol concentration in the oil with the inclusion of OMWW with fermented L. *plantarum* compared to the oil with plain/non-fermented OMWW, representing 703 and 112 mg/L oil, respectively. A similar pattern in this study was also observed for individual phenolic component, particularly oleuropein content, which represented 401.8 and 140.4 mg/L in oil samples with and without fermented *L. plantarum*, respectively. Furthermore, the isolated active molecules from OMWW such as hydroxytyrosol and oleuropein markedly found applications in the cosmetic industry [92].

The olive oil industry generates a great amount of olive pomace that can represent, on its own, considerable environmental/economic challenges. The re-utilization of this type of by-product is somewhat progressing in various applications. Examples are the commercial applications for direct uses such as edible vegetable oil and animal feed. The de-oiled fraction of pomace, namely, exhausted pomace, has found uses in agricultural applications including compost and soil amendment [93]. The de-fatted fraction also finds application in animal feed, commonly after being subjected to de-stoning [25], and pretreatment to decrease the lignin content [93], the high-molecular-weight polymers with water insolubility constituting roughly about 37% dry basis [97].

## 5. Conclusions and Future Prospects

In recent years, the distinctive value of olive oil products has been globally appreciated nutritionally and economically. However, the maintenance of the nutritive attributes is of great reliance on the fate the endogenous phenolics in the final product, which is highly dependent, favorably and adversely, on the types/parameters of processing and storage. To overcome the disadvantages inherent in malaxation parameters, researchers have proposed incorporation of auxiliary processing means based on green methodologies. However, given the challenges involved in gaining the increased yield of (i) desired phenolics and (ii) olive oil, together with other factors such as acceptability of organoleptic attributes, more extensive research work may be needed.

The intensification of nutritive quality of olive oil is of paramount importance, but this alone may not suffice in meeting the sustainability of the processing system in the olive oil industry. The huge generation of the biomass residues necessitates using efficient approaches to deal applicably with each type of waste stream generated during olive oil extraction which potentially enable (i) optimum waste management, particularly in the case of liquid effluent, and (ii) sustainable valorization of functional biomolecules. Olive mill by-products are often under-utilized—most have found low/moderate value applications while having an appreciably high added-value potential in various industrial sectors with great marketability. The mechanical processing techniques used for the extraction play decisively in the recovery of phenolics of interest. Researchers have made much effort to enable delivery of an ideal extraction system to optimally isolate target phenolics from the waste stream, which, together with other things, involves a great challenge due to the structural complexity of polyphenols and the inconsistent rhythms of their biosynthesis in various conditions. The inclusion of emerging techniques in the processing system has shown to be significantly effective in the increment of phenolic recovery, and there seem to be a movement towards adoption of these new technologies in some areas/applications. However, the prevalence of traditional/inefficient methods,

entails more research efforts to enable scalability of the extraction design, affordability, and simplicity of process operations.

Application of the green technologies in the olive oil industry, when the sustainable criteria are adequately fulfilled, helps achieve optimum production of olive oil and valorize the olive mill by-products that may (i) deal with the challenges associated with food/nutrition security, (ii) address the environmental issues, (iii) develop production/consumption of natural and healthy products, (iv) enable broader applications in the food/non-food system, and (v) improve marketability/investment return that can be of great value for the industrial system.

**Funding:** This review received no external funding.

**Conflicts of Interest:** The author declares no conflict of interest.

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
