# Peer review of "Effect of Processing on Phenolic Composition of Olive Oil Products and Olive Mill By-Products and Possibilities for Enhancement of Sustainable Processes"

_processes, doi:10.3390/pr9060953_

Round 1

Reviewer 1 Report

I suggest to change the type of Article from "Review" to "Perspective"

The authors should well marked the novelty character of this review.

This paragraph Olive Oil – An Overview of Characteristics of Typical Endogenous Phenolics and the Effect of Processing Conditions on their Loss/gain should be implemented and divided in subparagraphs and a Table with main extraction procedures and analytical techniques should be inserted.

The use of green analyichal techniques in olive and by-products should be better described and related references should be mentioned.

The linguistic revision of whole manuscript should be carried out.

The manuscript should be formatted by MDPI guidelines

Author Response

Dear Reviewer,

Many thanks for your helpful comments and taking the time to review my manuscript. The following list outlines a summary of the responses (in standard font, blue color) to the comments of the Reviewers (cited in italics):

Response to Reviewer 1 Comments:

“The authors should well marked the novelty character of this review.”

To present the significance/novelty of this review rather more clearly, the abstract has been re-drafted, also in the last part of the Introduction (page 2, the last paragraph), the description was re-drafted to elucidates the idea of the study more specifically.

May I also mention that this review presents an updated overview, with especial attentions to (i) the roles of processing factors in the fates of representative phenolics in both olive oil (main products) and olive olive mill by-products, from the perspective of the biosynthetic variabilities of each phenolic groups through various processing, and (ii) the sustainable alternatives (ideally incorporating emerging technologies) in the processing system to optimally manufacture nutritive-rich olive oil and valorize “high-value” uses of its by-products (rather than low/medium value ones).

This paragraph Olive Oil – An Overview of Characteristics of Typical Endogenous Phenolics and the Effect of Processing Conditions on their Loss/gain should be implemented and divided in subparagraphs … should be inserted”.

Revised accordingly.

Also, a table, highlighting the comparison of phase separation techniques (in both traditional and modern system) used to extract olive oil from the paste, has been added – Table 1. Page 5.

The use of green analytical techniques in olive and by-products should be better described and related references should be mentioned”.

Re-drafted accordingly.

“The linguistic revision of whole manuscript should be carried out”.

Double-checked & revised entirely.

“The manuscript should be formatted by MDPI guidelines”.

Revised and double-checked.

Reviewer 2 Report

This article presents an extensive review of the effects of processing on the phenolic composition of olive oils and their by-products. This review is well structured and presents useful information, but some information regarding the processing of the olive leaves is already presented by the author in the https://doi.org/10.3390/pr8091177. 
Also, the references need to be rechecked because there are many errors throughout the text. 
Lines 62 – 66 – This paragraph is hard to read. Please rewrite it. 
I recommend publishing this review after the author addresses the issues above.

Author Response

Dear Reviewer,

Many thanks for your helpful comments and taking the time to review my manuscript. The following list outlines a summary of the responses (in standard font, blue color) to the comments of the Reviewers (cited in italics):

Response to Reviewer 2 Comments:

This article presents an extensive review of the effects of processing on the phenolic composition of olive oils and their by-products. This review is well structured and presents useful information, but some information regarding the processing of the olive leaves is already presented by the author in the https://doi.org/10.3390/pr8091177”.

I do thank you for your kind comments.

Regarding the information related to the processing of olive leaves – very well-spotted, with thanks. The section of “olive leaves”, indeed constitutes a proportion, among other components of olive mill by-products, that was outlined in the designated sub-section in the present manuscript (rather than the expanded subject solely focusing on olive leaves that was reviewed in my previous paper). In this respect, the processing roles on phenolics, together with their potential use during olive oil extraction: Section 4, have been re-looked up in the literature and summarized in this manuscript, to, hopefully, make a proportional description, taking into consideration other types of the waste streams generated in olive oil industry. I do hope this would justify the reasoning.

Also, the references need to be rechecked because there are many errors throughout the text”.

The reference list/citations have been entirely revised and double-checked.

Lines 62 – 66 – This paragraph is hard to read. Please rewrite it”.

Thank you for pointing this out. As outlined to the previous comment, this section has been re-drafted and upgraded to explain the significance/idea of the review more clearly – it is now located in page 2. the last paragraph.

Reviewer 3 Report

The topic of the paper „Effect of Processing on Phenolic Composition of Olive Oil Products and Olive Mill By-Products and Possibilities for Enhancements of Sustainable Processes” is very interesting.

The introduction provides sufficient background and includes relevant references.

The manuscript is well written, and the text is easy to read.

The literature data are consistent and clearly presented.

The present manuscript discusses typical processing factors responsible for the fate of endogenous phenolics in olive oil products/by-products and provides an overview of the possibilities for the sustainable processing system to produce phenolic-rich olive oil, and optimally valorize the by-products.

In last years, the distinctive value of olive oil products has been globally appreciated nutritionally and economically.

The authors of the present manuscript show that by using conventional methods still exist largely in most industrial sectors, which entails more research efforts to enable scalability of the extraction design, affordability, and simplicity of process operation.

Observation:

At the reference number 35 the name of species Olea europaea is not italic.

Author Response

Dear Reviewer,

Many thanks for your helpful comments and taking the time to review my manuscript. The following list outlines a summary of the responses (in standard font, blue color) to the comments of the Reviewers (cited in italics):

Response to Reviewer 3 Comments:

The topic of the paper „Effect of Processing on Phenolic Composition of Olive Oil Products and Olive Mill By-Products and Possibilities for Enhancements of Sustainable Processes” is very interesting.

The introduction provides sufficient background and includes relevant references.

The manuscript is well written, and the text is easy to read.

The literature data are consistent and clearly presented.

The present manuscript discusses typical processing factors responsible for the fate of endogenous phenolics in olive oil products/by-products and provides an overview of the possibilities for the sustainable processing system to produce phenolic-rich olive oil, and optimally valorize the by-products.

In last years, the distinctive value of olive oil products has been globally appreciated nutritionally and economically.

The authors of the present manuscript show that by using conventional methods still exist largely in most industrial sectors, which entails more research efforts to enable scalability of the extraction design, affordability, and simplicity of process operation.

Observation:

At the reference number 35 the name of species Olea europaea is not italic.

Many thanks for your great comments.

The above observations have been revised. The term “Olea europaea” was re-amended to italics – is now located in Ref. No. 36.

Round 2

Reviewer 2 Report

All the suggested modifications were made by the author and the manuscript may be accepted in present form.